# Flat acoustics with soft gradient-index metasurfaces

Yabin Jin[1], Raj Kumar[2], Olivier Poncelet[1], Olivier Mondain-Monval[2] & Thomas Brunet[1]

Recently, metasurfaces have been proven to be effective and compact devices for the design of arbitrary wavefronts. Metasurfaces are planar metamaterials with a subwavelength thickness that allows wavefront shaping by introducing in-plane variations, namely, gradients, in the spatial wave response of these flat structures. Here we report a new class of acoustic gradient-index (GRIN) metasurfaces engineered from soft graded-porous silicone rubber with a high acoustic index for broadband ultrasonic three-dimensional wavefront shaping in water. The functionalities of these soft flat lenses are illustrated through various experiments, which demonstrate beam steering and beam focusing, as well as vortex beam generation in free space. These new GRIN metasurfaces may have important applications in various domains using designed ultrasonic fields (biomedical imaging, industrial non-destructive testing, contactless particle manipulation), since their fabrication is very straightforward with common polymer science engineering.

---

[1] Univ. Bordeaux, CNRS, Bordeaux INP, ENSAM, UMR 5295 I2M, F-33405 Talence, France. [2] Univ. Bordeaux, CNRS, UMR 5031 CRPP, F-33600 Pessac, France. These authors contributed equally: Yabin Jin, Raj Kumar. Correspondence and requests for materials should be addressed to T.B. (email: thomas.brunet@u-bordeaux.fr)

As an alternative to 3D metamaterials[1–3], the recent emergence of metasurfaces[4–6], has introduced new opportunities to realize wavefront shaping without restriction (e.g., directivity design, focusing, and singular field generation) by introducing in-plane-varying phase discontinuities across the metasurface. These abrupt phase shifts are usually induced by scatterers dispersed in 2D arrays with some specific patterns depending on the targeted application. However, the phase response of these decorated membranes is usually in a narrow frequency band depending on the size and properties of the particles. Alternatively, local phase shifts may be induced by greatly increasing the wave path length $nd$ (where $n$ is the refractive index and $d$ is the thickness of the material) while keeping subwavelength thicknesses with respect to the background medium. In acoustics, the concept of coiling space[7–9] follows this idea by artificially enlarging the airborne propagation distance $d$ through tapered labyrinthine unit cells to allow for large phase shifts. This approach is interesting, since these structures can be easily fabricated using 3D printing and may be quantized with coding bricks[10,11]. However, such a design for wavefront modulation usually works for a rather narrow selected frequency band and is inappropriate for manipulating acoustic

waves that are propagating in a high-impedance media, such as waterborne waves.

Here, we propose to overcome these problems thanks to another approach that consists of increasing the acoustic path length $nd$ by using high-refractive-index materials with weak dispersion properties. Such a gradient-index (GRIN) metasurface requires materials with indices that are both high and tuneable over a broad range of values. Recently, soft porous silicone rubber materials[12,13] have been shown to exhibit a wide range of tuneable acoustic indices $n$ (from 1.5 to 25 with respect to water[14]) depending on the material porosity $\Phi$. In this work we propose to utilize this great versatility to design a new class of soft porous GRIN metasurfaces for shaping acoustic wavefronts radiating in 3D open space.

## Results

**Soft porous silicone rubbers with highly controlled porosities.** To accurately control the value of the acoustic index $n$ in the soft porous materials used to build GRIN metasurfaces, we employed an emulsion templating method that was developed in previous works[15,16], coupled with a supercritical drying technique[17] to avoid pore collapse during the drying process[18]. Thus, our porous

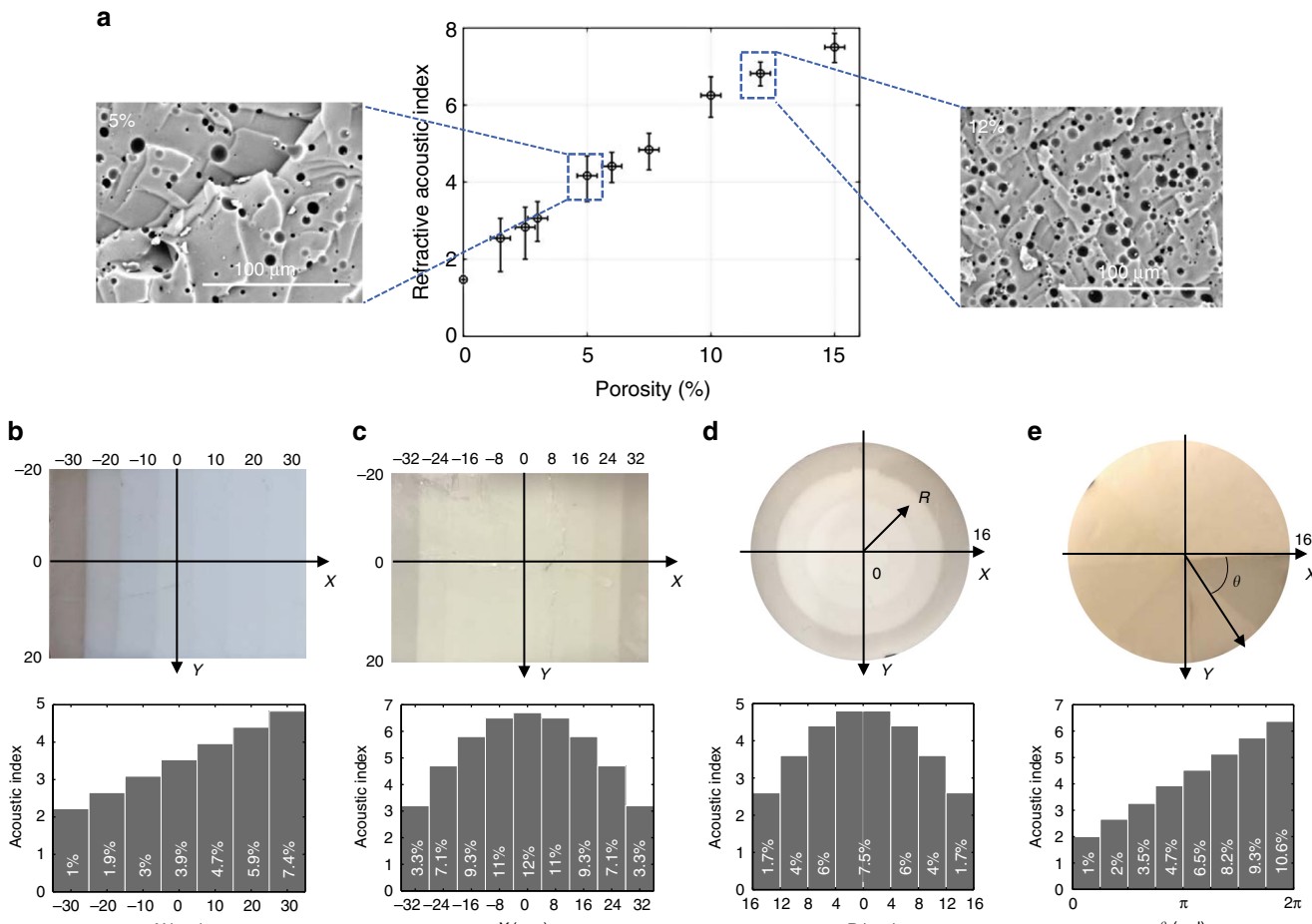

**Fig. 1** Soft gradient-index metasurfaces with highly controlled gradients of porosity. **a** Measured relative refractive acoustic index $n$ for various soft porous silicone rubbers with porosities ranging from 0% to 15%. Insets: Scanning electron microscopy images of the porous structure for $\Phi = 5\%$ and 12%; These soft porous materials are fabricated with a targeted porosity controlled with an accuracy of 1%. **b–e** Examples of soft porous GRIN metasurfaces made of soft porous silicone rubber stripes with a thickness $d = 2$ mm. The corresponding acoustic index profiles are shown in the histograms below. The values mentioned at the bottom of the histograms correspond to the porosity in each porous stripe. The linear index profile shown in **b** is designed for beam deflection with an angle of 5°, while the hyperbolic index profile shown in **c** is designed for beam focusing with a focal length $F = 70$ mm. The radial index profile shown in **d** is designed for 3D sound focusing with a focal length $F = 20$ mm, while the azimuthal index profile shown in **e** is designed for the generation of an acoustic vortex beam with a topological charge $l = +1$

materials are obtained with a targeted porosity having an accuracy of 1%. As an example, various soft porous silicone rubber materials were synthesized with different porosities $\Phi$ and were acoustically characterized, providing materials with a relative refractive acoustic index $n$ ranging from 1.4 ($\Phi = 0\%$) to 7.5 ($\Phi = 15\%$), as shown in Fig. 1a. The speed of sound does not depend significantly on the frequency in these soft porous materials as previously reported[14]; therefore, that the real part of the acoustic index remains constant within a broad frequency range (see Supplementary Figure 2). As a result, broadband soft GRIN metasurfaces can be fabricated with any desired index profile by spatially assembling different units with a perfectly controlled porosity.

**Wavefront shaping with unidirectional gradients of index.** For clarity, we first consider metasurfaces having a gradient of index along only one direction, taken here to be along the $X$-axis, to demonstrate the ability of our soft porous metasurfaces to shape planar wavefronts. According to the generalized Snell's law[4], the index profile must be linear to deflect a plane wave (Eq. (1)), while beam focusing requires a hyperbolic gradient of the following index (Eq. (2)):

$$n(X) = n(X = 0) + \frac{\sin(\theta)X}{d} \quad (1)$$

$$n(X) = n(X = 0) - \frac{\sqrt{X^2 + F^2} - F}{d} \quad (2)$$

In Eq. (1), $\theta$ is the angle of deflection, and in Eq. (2), $F$ is the targeted focal length for beam focusing. Two large rectangular samples were fabricated (Fig. 1b, c) with a subwavelength thickness ($d = 2$ mm). These two metasurfaces are made of strip units with varying porosities along the $X$-axis to obtain a gradient of the index along that direction, with the acoustic index $n$ being constant along the $Y$-axis. From Eqs. (1) and (2) and by considering the relation $n(\Phi)$ given in Fig. 1a, the first sample (Fig. 1b) was designed to deflect an incident acoustic plane wave with an angle of 5°, while the second sample (Fig. 1c) was designed to focus sound with a focal length $F$ of 70 mm.

To demonstrate the deflection (or focalization) of an acoustic plane wavefront by our soft porous GRIN metasurfaces, we covered a large planar ultrasonic transducer with a linear (or hyperbolic) GRIN sample, as shown in Fig. 2a. The central frequency of the transducer used here is 150 kHz, corresponding to $\lambda_0 = 10$ mm for waterborne waves, i.e., five times larger than the sample thickness $d$ (=2 mm). Then, the pressure field radiated from these subwavelength materials was scanned in the $XZ$ plane, with the $Z$-axis being the incident plane wave direction, using a tiny hydrophone having a diameter of 1 mm. On the one hand, Fig. 2c shows a clear deflection of the reference acoustic beam (shown in Fig. 2b) with an angle of 5°, as targeted by the theoretical design. On the other hand, the pressure-field map shown in Fig. 2d reveals clear focusing effects with a focal length $F$ of ~50 mm that slightly differs from the targeted theoretical focal length (=70 mm). This difference is due to the small Fresnel number of our flat lens, which induces a focal-shift effect, as previously reported in optics[19]. To quantitatively analyze these observations, we also performed finite element simulations (Fig. 2e) revealing excellent agreement with experiments (Fig. 2f); experiments at 90 kHz clearly demonstrate the subwavelength functionality of our soft metasurface ($d < \lambda_0/8$ at this frequency in water). From the amplitude profiles extracted at $X = 0$ mm and $Z = 44$ mm, the size of the focal spot was fully characterized, exhibiting full widths at half maximum of $4.9\lambda_0$ and $0.8\lambda_0$ along the $Z$-axis (Fig. 2g) and $X$-axis (Fig. 2h), respectively.

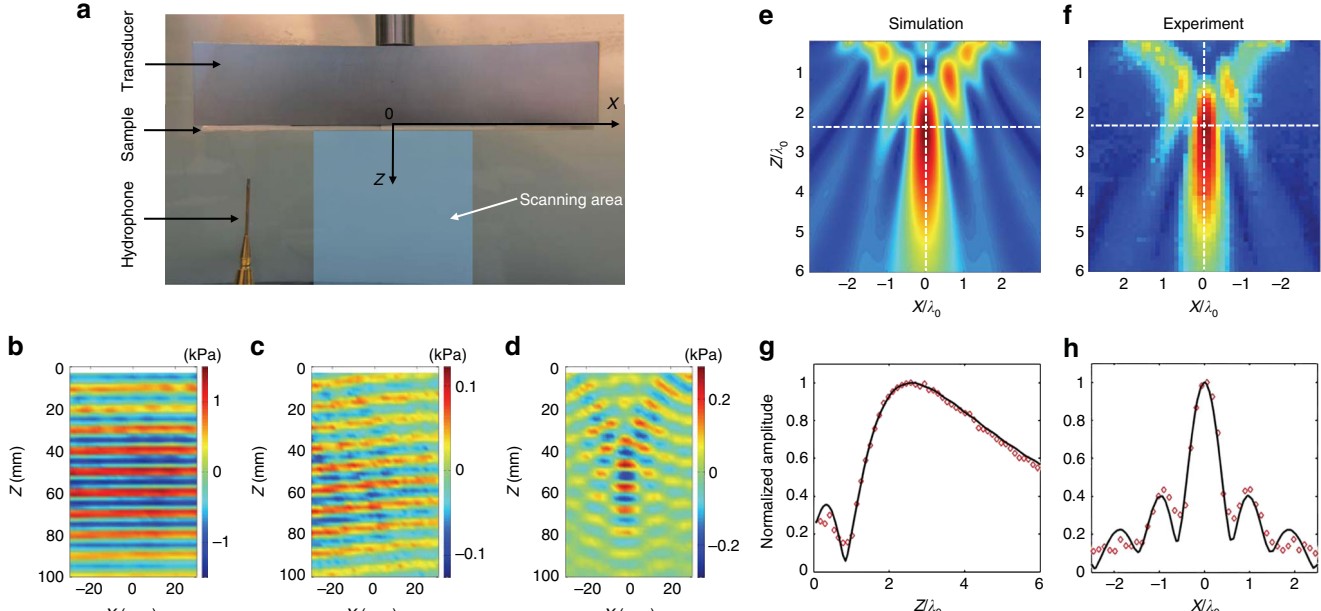

**Fig. 2** 2D deflecting and focusing of ultrasound. **a** Experimental setup showing a soft porous GRIN metasurface covering a large rectangular ultrasonic transducer immersed in a large water tank. A tiny hydrophone is mounted on motorized linear stages to scan the acoustic pressure field in the $XZ$ plane (60 mm × 100 mm) emerging from the metasurface (shaded blue area). **b–d** Snapshots of the measured pressure fields shown at a given time without any sample (**b**), with the deflecting metasurface (**c**), and with the focusing metasurface (**d**) (corresponding videos are shown in Supplementary Movies 1, 2 and 3). Simulated (**e**) and measured (**f**) acoustic field patterns extracted from fast Fourier transforms performed at 90 kHz on the signals recorded for the focusing metasurface (**d**). Measured (red diamonds) and simulated (black line) normalized amplitude field distributions along the $Z$-axis for $X = 0$ mm (**g**) and along the $X$-axis for $Z = 44$ mm (**h**). The full widths at half maximum peaks are $4.9\lambda_0$ and $0.8\lambda_0$ along the $Z$-axis and $X$-axis, respectively

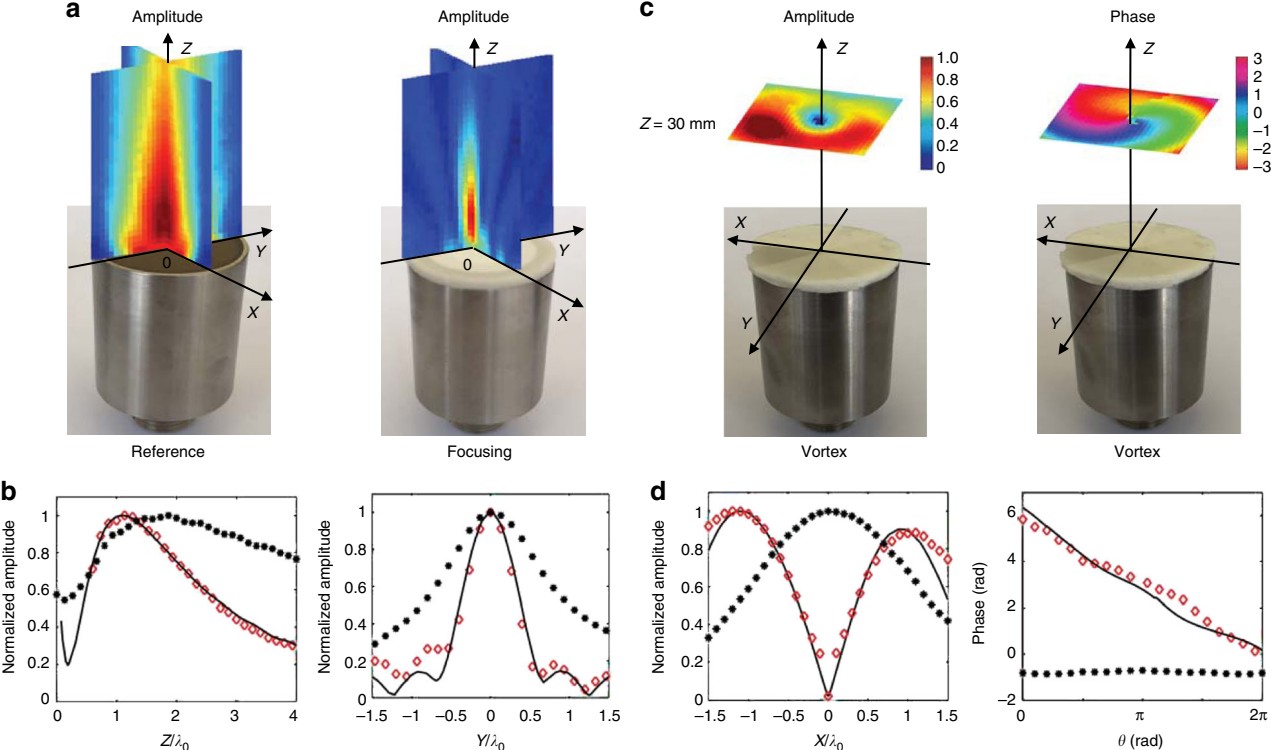

**Fig. 3** 3D focusing of ultrasound and acoustic vortex generation. **a** Acoustic field patterns measured at 200 kHz in the $XZ$ and $YZ$ planes (30 mm × 30 mm) without (reference) and with (focusing) the circular radial GRIN metasurface (Fig. 1d) deposited on a 1-inch-diameter ultrasonic immersion transducer. **b** Normalized amplitude field distributions along the $Z$-axis for $X = 0$ mm and $Y = 0$ mm (left) and along the $Y$-axis for $Z = 9$ mm and $X = 0$ mm (right). The full widths at half maximum peaks are $2.4\lambda_0$ and $0.8\lambda_0$ along the $Z$-axis and $Y$-axis, respectively. **c** Amplitude and phase of the acoustic field measured at 150 kHz in the $XY$ plane (30 mm × 30 mm) at $Z = 30$ mm with the circular azimuthal GRIN metasurface (Fig. 1e) deposited on the same transducer as in **a**. All the key features of an acoustic vortex are evident here: the amplitude of the wave field is null at the center, while the phase is winding linearly along that point. **d** Normalized amplitude and phase field distributions along the $X$-axis and the angle $\theta$, respectively. For all distributions displayed here, open red symbols refer to measurements with the metasurface, while filled black symbols correspond to reference measurements without the sample. Solid lines refer to numerical simulations obtained with the metasurfaces

**3D focusing of ultrasound with radially graded flat lens.** Now that the functionalities of our soft porous GRIN metasurfaces have been demonstrated with a gradient of the index along one direction, we also designed circular metasurfaces with a 2D gradient of the index that may be used as flat lenses for shaping the wavefront emitted by standard commercial immersion ultrasonic transducers. As a first example, we fabricated a 32-mm-diameter circular metasurface with a radial gradient index (Fig. 1d) to focus sound in water by simply depositing the sample on a 1-inch-diameter transducer (UltranGroup GS200-D25) having a central frequency of 200 kHz ($\lambda_0 = 7.5$ mm in water). The transmitted acoustic beam was experimentally characterized with the same tiny hydrophone as used previously (Fig. 2a). As shown in Fig. 3a, the metasurface induces clear focusing effects in comparison to that of the uncoated transducer. As predicted by numerical simulations (Fig. 3b), the focal length $F$ was 9 mm, corresponding to a numerical aperture (NA) of 0.82, with the maximal half-angle of the acoustic cone being 55°. Our metasurface exhibits a considerably larger NA compared to the numerous bulky acoustic focusing flat lenses recently tested in water[20–22]. However, an acoustic Fresnel lens[23] with a similar NA was also reported, but its focal length is considerably larger ($F = 15\lambda_0$) than that of our soft meta-surface ($F = 1.2\lambda_0$). Therefore, our flat focusing lens might be used for acoustic microscopy given its high numerical aperture and its low-amplitude side lobes, as shown in Fig. 3b.

**Acoustic vortex generation with azimuthally graded flat lens.** In wave physics, it is also of prime importance to generate more complex wavefields, such as vortex beams, that exhibit phase singularity along their propagation axis. These helical beams are now commonly used in the field of optical manipulation since the optical vortex beams have been demonstrated to act like optical tweezers[24]. In acoustics, vortex generation is now fully reliable and accurate for manipulating small objects in air[25] and water[26]. In most cases, large arrays of transducers associated with sophisticated electronics are required[27], although some compact acoustic metamaterials have been recently reported for the generation of acoustic vortex beams[28,29]. In addition, acoustic metasurfaces may also provide new insights[30]. Therefore, we designed a circular porous sample with the same dimensions as the previous flat focusing lens but exhibiting an azimuthal gradient of the index (Fig. 1e). The angular variations of the output phase induced by this GRIN metasurface are responsible for the generation of a helical beam, as shown in Fig. 3c. The non-uniformity of the ring amplitude is due to the impedance mismatch between the porous sample and the surrounding water, which depends on the material porosity (see Supplementary Figure 1). Nevertheless, all the key features associated with the vortex beams are quantitatively demonstrated in Fig. 3d since the amplitude of the wave field is null in its center around which the phase is winding linearly. The topological charge $l$ of the acoustic vortex

shown here is +1 since the phase achieves only one jump of $2\pi$ on a close contour around the phase singularity. Thus, we demonstrate that an acoustic vortex can be generated in free space with only a single transducer coated with our soft metasurface.

## Discussion

In conclusion, we report a new approach for creating various types of acoustic wavefronts in free space by using soft gradient-index porous metasurfaces. From planar isophase sources, these flat acoustic lenses were shown to generate steered planar, focused spherical and helical acoustic wavefronts. Although focused and helical beam generation is the most attracting application, the scope of possibilities for these metasurfaces extends further, including the graded-index profile of the metasurface being able to produce any phase distribution and thus acting as a subwavelength phase mask. Additionally, whereas most of the recent reported metasurfaces are designed for a single frequency dedicated to a specific application, we demonstrate here that our acoustic metasurfaces can have various broadband functionalities (e.g., beam steering, sound focusing, vortex beam generation) by manipulating the spatial distribution of the material porosity. The fabrication processes of these metasurfaces rely on common soft-matter techniques that allow for possible up-scaling and low-cost production. Thus, these new acoustic flat lenses may open a new route for many applications, such as acoustic microscopy (e.g., for non-destructive testing), for which a high numerical aperture is required, and 3D contactless manipulation of micro-objects with acoustic vortex beams.

## Methods

**Experiments**. The 10-cycle tone burst input Gaussian pulse is generated using a waveform generator (Agilent 33500B Series) with an 8-V input voltage to the piezoelectric ultrasonic transducers (Imasonic and Olympus). A 1-mm-diameter hydrophone (Precision Acoustics) is used to record acoustic signals in the scanning areas using motorized linear stages (Newport). The signals are acquired via a digitizer card (AlazarTech ATS460) on a computer.

**Simulation**. The finite element software COMSOL Multiphysics® is used for simulation with the acoustic–solid interaction model. The acoustic velocity of the metasurface is defined as $c = \omega/k$, where $\omega$ is the angular frequency and the complex wavenumber is $k = \omega/c - i\alpha$ (with the $e^{i\omega t}$ convention). The attenuation factor is considered in the form $\alpha = \alpha_0 f^{1.5}$, where the unit of frequency $f$ is MHz.

## Data availability

The data that support the findings of this study are available from the corresponding author upon reasonable request.

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

## Acknowledgements

We thank Tania Ireland and Frédéric Marchal from Elkem Silicones Company for productive discussions and for providing us with the silicone rubber. We thank Jacques Leng, Samuel Marre, and Artem Kovalenko for productive discussions and advices. This work was partially funded and performed within the framework of the Labex AMADEUS ANR-10-LABEX-0042-AMADEUS with the help of the French state Initiative d'Excellence IdEx ANR-10-IDEX-003-02 and project BRENNUS ANR-15-CE08-0024 (ANR and FRAE funds).

## Author contributions

T.B. conceived the original idea and O.M.M. provided the original materials. T.B., O.M.M., and O.P. supervised the project. R.K. fabricated the samples under the guidance of O.M.M. Y.J. performed the numerical work. Y.J. and T.B. performed the experiments. Y.J., T.B. and O.P. analyzed the data and wrote the manuscript. All authors discussed the results and commented on the manuscript.

## Additional information

**Competing interests:** The authors declare no competing interests.

