## [Peer Review File · Nature Communications]

Reviewers' Comments:

Reviewer #1:

Remarks to the Author:

In this paper, the author reported a flat acoustic gradient-index metasurface made from soft graded-porous silicone rubber. This flat metasurface is demonstrated to generate e.g. steered planar, focused spherical and helical acoustic wave-fronts, all from planar isophase sources.

It is overall a good work, the material is interesting, the analysis in the manuscript have been carried out thoroughly and the results are scientifically sound. However, metasurface is already a relatively mature concept and has a lot of related works even in acoustics. Although this flat and soft acoustic metasurface is very practical, the paper does not have much physical novelty. After careful consideration and much thought, I am very sorry to say I disagree with its publication in Nature Communications.

Reviewer #2:

Remarks to the Author:

The manuscript presents the experimental demonstration of several metamaterial lenses made of porous silicone rubber. The technology behind these engineered materials has been perfected by the authors (see Refs. 12, 13, 16) to the point where the porosity level can be tightly controlled inside a macroscopic sample (i.e. only 1% error in porosity size is reported). This provides a new method to manipulate sound speed in water. The acquired ability to engineer sound speed is convincingly demonstrated through the implementation of several gradient index devices. Among them, the lens that produces vortex beams is the most complicated and illustrates very well the level of control provided by the porous silicone. Given how notoriously difficult is to control underwater sound, I believe this manuscript will be of interest to a large audience and deserves publication in Nature Communications. There are however several issues that should be addressed before publication.

- A major issue is that of insertion loss. The soft metamaterial lenses fabricated cover the face of an underwater transducer. I expect the pores (are they trapped air bubbles?) to strongly scatter the sound produced by the transducer. As a result, the insertion loss should be significant (see Figs. 2c & 2d). The manuscript should quantify the insertion loss (e.g. by reporting the lens impedance).
- Fig. 3c shows a drastic amplitude decrease in a region in which high phase shifts of π - π are required, which suggest that the impedance varies significantly with the refractive index (and with the porosity density). The manuscript should also report the variation of metamaterial impedance versus refractive index.
- Fig 1b shows the quasi-plane wave produced by the bare transducer. Why is the pressure amplitude higher in the $30\text{ mm} < z < 80\text{ mm}$ region than closer to the transducer ($z < 30\text{ mm}$)?
- The introduction states that the labyrinthine unit cells "usually work for a narrow selected frequency band". This statement is inaccurate. The labyrinthine cells have relatively low dispersion, and more importantly the gradient of n is almost constant (Xie et al, APL 103, 201906, 2013), which means that lenses made of these metamaterials are typically broadband.
- The manuscript should report the dispersion characteristics of the porous rubber silicone used. How do the refractive index and impedance vary with frequency?
- What is the resonant frequency of the porosities?
- The authors call their device a metasurface. I strongly suggest against using the term

metasurface here. Unlike other metasurfaces demonstrated in the past which are less than 4 unit cells thick, the fabricated lenses are many dozens of unit cells in the propagation direction (see Fig. 1a).

- Vortex beams have been demonstrated in passive devices. The manuscript should acknowledge past work, e.g. (Naify et al, APL 108, 223503, 2016) and others.

Reviewers' comments:

Reviewer #1 (Remarks to the Author):

In this paper, the author reported a flat acoustic gradient-index metasurface made from soft graded-porous silicone rubber. This flat metasurface is demonstrated to generate e.g. steered planar, focused spherical and helical acoustic wave-fronts, all from planar isophase sources.

It is overall a good work, the material is interesting, the analysis in the manuscript have been carried out thoroughly and the results are scientifically sound. However, metasurface is already a relatively mature concept and has a lot of related works even in acoustics. Although this flat and soft acoustic metasurface is very practical, the paper does not have much physical novelty. After careful consideration and much thought, I am very sorry to say I disagree with its publication in Nature Communications.

Reviewer #2 (Remarks to the Author):

The manuscript presents the experimental demonstration of several metamaterial lenses made of porous silicone rubber. The technology behind these engineered materials has been perfected by the authors (see Refs. 12, 13, 16) to the point where the porosity level can be tightly controlled inside a macroscopic sample (i.e. only 1% error in porosity size is reported). This provides a new method to manipulate sound speed in water. The acquired ability to engineer sound speed is convincingly demonstrated through the implementation of several gradient index devices. Among them, the lens that produces vortex beams is the most complicated and illustrates very well the level of control provided by the porous silicone. Given how notoriously difficult is to control underwater sound, I believe this manuscript will be of interest to a large audience and deserves publication in Nature Communications. There are however several issues that should be addressed before publication.

- A major issue is that of insertion loss. The soft metamaterial lenses fabricated cover the face of an underwater transducer. I expect the pores (are they trapped air bubbles?) to strongly scatter the sound produced by the transducer. As a result, the insertion loss should be significant (see Figs. 2c & 2d). The manuscript should quantify the insertion loss (e.g. by reporting the lens impedance).

- Fig. 3c shows a drastic amplitude decrease in a region in which high phase shifts of π/π are required, which suggest that the impedance varies significantly with the refractive index (and with the porosity density). The

manuscript should also report the variation of metamaterial impedance versus refractive index.

- Fig 1b shows the quasi-plane wave produced by the bare transducer. Why is the pressure amplitude higher in the $30 \text{ mm} < z < 80 \text{ mm}$ region than closer to the transducer ($z < 30 \text{ mm}$)?

- The introduction states that the labyrinthine unit cells "usually work for a narrow selected frequency band". This statement is inaccurate. The labyrinthine cells have relatively low dispersion, and more importantly the gradient of n is almost constant (Xie et al, APL 103, 201906, 2013), which means that lenses made of these metamaterials are typically broadband.

- The manuscript should report the dispersion characteristics of the porous rubber silicone used. How do the refractive index and impedance vary with frequency?

- What is the resonant frequency of the porosities?

- The authors call their device a metasurface. I strongly suggest against using the term metasurface here. Unlike other metasurfaces demonstrated in the past which are less than 4 unit cells thick, the fabricated lenses are many dozens of unit cells in the propagation direction (see Fig. 1a).

- Vortex beams have been demonstrated in passive devices. The manuscript should acknowledge past work, e.g. (Naify et al, APL 108, 223503, 2016) and others.

** See Nature Research's author and referees' website at www.nature.com/authors for information about policies, services and author benefits

Authors: First, we sincerely thank the two reviewers for their careful reading of the manuscript and their constructive comments. Below, we answer all their questions. Information related to the main issues raised by the two referees has been detailed in the Supplementary file.

Reviewer #1: *“In this paper, the author reported a flat acoustic gradient-index metasurface made from soft graded-porous silicone rubber. This flat metasurface is demonstrated to generate e.g. steered planar, focused spherical and helical acoustic wavefronts, all from planar isophase sources.*

It is overall a good work, the material is interesting, the analysis in the manuscript have been carried out thoroughly and the results are scientifically sound. However, metasurface is already a relatively mature concept and has a lot of related works even in acoustics. Although this flat and soft acoustic metasurface is very practical, the paper does not have much physical novelty. After careful consideration and much thought, I am very sorry to say I disagree with its publication in Nature Communications”.

Authors: We first thank the reviewer for his/her interest in our work. However, we cannot agree with his/her statement regarding the “physical novelty” for the following reasons. The field of “metasurfaces” has been receiving significantly increasing attention since the seminal work by Yu *et al.* [*Science* **334**, 333, 2011]. This concept was extended to acoustic waves by Xie *et al.* [*Nature Communications* **5**, 5553, 2014] and Li *et al.* [*Physical Review App.* **2**, 064002, 2014] a few years later (see Refs. 7-9). Following the concept of the generalized Snell’s law, most published works have addressed manipulation of air-borne waves with coiling-space/labyrinthine structures, which involves artificially increasing the propagation distance L to extend the wave path kL (where k is the wavenumber). However, these devices cannot be applied for wave propagation in high-impedance medium, e.g., water, although many applications are largely demanded in underwater acoustics. In our work, we propose a new and more versatile approach to shape an incoming water-borne acoustic plane wave into any wavefront at will (cylindrical, spherical, helical and so on) by using broadband porous gradient-index metasurfaces. The demonstration is carried out through various ultrasonic experiments showing deflecting and focusing effects as well as the generation of complex wave-fields such as singular waves, or so-called vortex beams. Since our metasurfaces are compact and very simple to use compared to common acoustic devices (bulky lenses, large arrays of transducers), these flat acoustic lenses may be useful for many applications such as acoustic microscopy/imaging or contactless manipulation of small particles with acoustic vortex beams. Notably, all these ultrathin porous materials are relatively easy to generate from common polymer science processes, thus providing a new class of flat, low-cost acoustic devices for wavefront shaping.

Reviewer #2: *“The manuscript presents the experimental demonstration of several metamaterial lenses made of porous silicone rubber. The technology behind these engineered materials has been perfected by the authors (see Refs. 12, 13, 16) to the point where the porosity level can be tightly controlled inside a macroscopic sample (i.e. only 1% error in porosity size is reported). This provides a new method to manipulate sound speed in water. The acquired ability to engineer sound speed is convincingly demonstrated through the implementation of several gradient index devices. Among them, the lens that produces vortex beams is the most complicated and illustrates very well the level of control provided by the porous silicone. Given how notoriously difficult is to control underwater sound, I believe this manuscript will be of interest to a large audience and deserves publication in Nature Communications. There are however several issues that should be addressed before publication”.*

Authors: We thank the reviewer for his/her careful reading and kind recommendation to publish our manuscript in *Nature Communications*.

Q1: *“A major issue is that of insertion loss. The soft metamaterial lenses fabricated cover the face of an underwater transducer. I expect the pores (are they trapped air bubbles?) to strongly scatter the sound produced by the transducer. As a result, the insertion loss should be significant (see Figs. 2c & 2d). The manuscript should quantify the insertion loss (e.g. by reporting the lens impedance)”.*

Authors: Insertion loss is a very important issue and we thank the reviewer for giving us an opportunity to clarify this point. First, the pores are not “air bubbles” but rather “air cavities” since the porous material is composed of a solid visco-elastic soft rubber and not a liquid-like medium such as bubbly media. The size of our air cavities (1-10 μm , see the SEM images in Fig. 1a) is much smaller than the typical acoustic wavelengths considered in this study that are greater than 1 mm. Consequently, the pores do not strongly scatter the sound of the transducer but only increase the compressibility of the porous material with a very low sound speed. Notably, the intrinsic absorption in these porous materials is larger than the absorption in the non-porous material as previously observed in similar soft porous silicone rubbers [see Ba *et al. Scientific Reports* 7, 40106, 2017]. However, insertion losses are mainly due to the impedance mismatch between the metasurfaces and the surrounding water since our soft porous metasurfaces are very thin.

As suggested by the referee, we added a new figure in the Supplementary file (Fig. S1) showing the lens impedance Z for all the metasurfaces reported in this study. Note that the impedance Z shown in Fig. S1 has been defined as the product of the mass density multiplied by the sound speed of each sector, which is the definition for non-absorbing materials. Here, taking into account complex-valued acoustic parameters in the definition of Z does not have strong impact for gauging the impedance mismatch (see detailed discussion in the Supplementary file).

We also added one sentence in the main text (see the last paragraph before the conclusion): *“The non-uniformity of the ring amplitude [...], which depends on the material porosity (see Fig. S1 in the Supplementary file).”*

Q2: “Fig. 3c shows a drastic amplitude decrease in a region in which high phase shifts of π/π are required, which suggest that the impedance varies significantly with the refractive index (and with the porosity density). The manuscript should also report the variation of metamaterial impedance versus refractive index.”

Authors: The minimum amplitude in the vortex ring shown in Fig. 3c is observed in front of the highest porous sector of the metasurface. Indeed, the impedance mismatch between the metasurface and the surrounding water is maximal in this region such that the transmitted field is minimal. As mentioned in the previous comment (Q1), we now provide new quantitative data regarding the impedance of each sample in the new Supplementary file (Fig. S1). To complete this information, the variation of the metamaterial impedance versus the refractive index is shown below as required by the reviewer:

Q3- “Fig 1b shows the quasi-plane wave produced by the bare transducer. Why is the pressure amplitude higher in the $30\text{ mm} < z < 80\text{ mm}$ region than closer to the transducer ($z < 30\text{ mm}$)?”

Authors: A ten-cycle tone burst excitation is applied to the emitting transducer in all experiments, which generates a Gaussian acoustic pulse propagating from the transducer to the bottom of the water tank as shown in Fig. 1a. The snapshots that are displayed in Figs. 2b-d show the pressure field recorded in the scanned area at a given time when the 10-cycle acoustic pulse is located in the $30\text{-mm} < z < 80\text{-mm}$ region. The amplitude decrease in the $0 < z < 30\text{-mm}$ region is the tail of the 10-cycle pulse. The full videos showing the propagation of these Gaussian pulses from the top to the bottom of the water tank are given in the Supplementary file. We also slightly changed the caption of Fig. 2 for clarification.

Q4- “The introduction states that the labyrinthine unit cells “usually work for a narrow selected frequency band”. This statement is inaccurate. The labyrinthine cells have relatively low dispersion, and more importantly the gradient of n is almost constant (Xie *et al.*, *APL* 103, 201906, 2013), which means that lenses made of these metamaterials are typically broadband”.

Authors: The reviewer refers to previous studies addressing “space-coiling acoustic metamaterials” conducted by Cummer’s group. The authors reported the measurement of a broadband negative index [Xie *et al.*, *Phys. Rev. Lett.* **110**, 175501 (2013)] and also numerically studied five space-coiling acoustic metamaterial unit cell designs [Xie *et al.*, *Appl. Phys. Lett.* **103**, 201906 (2013)] for broadband impedance matching. In these cases, the term “broadband” indicates that the real part n' of the complex-valued acoustic index $n (=n'+in)$ has negative values over broad frequency ranges, which is very interesting and undisputable. However, the real part n' of the acoustic index n strongly depends (linearly) on the frequency regardless of the unit cell, thus inducing high dispersion effects in these acoustic metamaterials.

On the other hand, the unit cells of our metasurfaces, which are pieces of porous rubber, are non-dispersive since the sound speeds (or the real part of the acoustic index) are quasi constant with frequency (see Fig. S2 in the Supplementary file). In our paper, the term “broadband” refers to acoustic properties that are not frequency dependent.

Q5- “The manuscript should report the dispersion characteristics of the porous rubber silicone used. How do the refractive index and impedance vary with frequency?”

Authors: As suggested by the reviewer, the frequency-dependence of both the refractive acoustic index and acoustic impedance has been provided in the Supplementary file. As shown in Fig. S2, these two parameters are quasi constant with the frequency, demonstrating that the dispersion in our porous materials is very low. Thus, our soft porous metasurfaces can be considered broadband acoustic devices.

Q6- “What is the resonant frequency of the porosities?”

Authors: As reminded by Leroy *et al.* [*Applied Physics Letters* **95**, 171904, 2009], the Minnaert resonant frequency ω_0 of an air cavity embedded in a soft elastomeric matrix

(*e.g.*, soft silicone rubber here) is given by $\omega_0 = \frac{1}{R} \sqrt{\frac{3\beta_{air} + 4\mu}{\rho}}$, where β_{air} (= 0.13

MPa) is the longitudinal modulus of air, μ (~ 1 MPa) is the shear modulus, ρ (= 1020 kg/m³) is the mass density of the elastomeric matrix, and R is the radius of the air cavity. Since the pore size of our porous materials is between 1 and 10 μm , the resonance frequency of these air cavities is larger than 3 MHz, which is at least 10-times higher than the operating frequencies considered in our study (100 - 200 kHz). As a result, the porous silicone rubber material that we used in our work is not a locally resonant bulk material in the frequency range considered here.

Q7- “The authors call their device a metasurface. I strongly suggest against using the term metasurface here. Unlike other metasurfaces demonstrated in the past which are less than 4 unit cells thick, the fabricated lenses are many dozens of unit cells in the propagation direction (see Fig. 1a).”

Authors: By definition, a metasurface refers to an artificial sheet material with *sub-wavelength thickness*, indicating that the metasurface thickness must be smaller than the incident wavelength while still strongly modulating the behaviour of the incident waves through specific conditions. All our metasurfaces are 2 mm thick, which is 5-times smaller than the typical incident wavelength λ_0 that we considered in our study ($\lambda_0 = 10$ mm in water at 150 kHz for ultrasound). Thus, we believe that the term ‘metasurface’ is fully appropriate in our case.

The sentence “*the fabricated lenses are many dozens of unit cells in the propagation direction*” is somewhat confusing to us. What does the reviewer refer to by saying “*dozens of unit cells*”? He/she perhaps assumes that the air-filled pores are elementary units, which is not correct. The elementary units that constitute our soft porous metasurfaces are small strips composed of porous materials with sub-wavelength thickness. As shown in Fig. 1b-c, the lateral dimensions of these strips are approximately 10 mm x 40 mm. By assembling some of these porous strips with varying porosities, we create soft gradient-index metasurfaces as shown in Fig. 1b-c.

Q8- “Vortex beams have been demonstrated in passive devices. The manuscript should acknowledge past work, e.g. (Naify et al, APL 108, 223503, 2016) and others.”

Authors: We thank the reviewer for drawing our attention to these papers. We added some references in the main text addressing the generation of acoustic vortex beams (see new Refs. 27-30 in the main text).

Reviewers' Comments:

Reviewer #1:

Remarks to the Author:

According to the reply, the authors claim their novelties are: (1) the previous acoustic metasurfaces based on generalized Snell's law's cannot be used for underwater sound, but this proposed acoustic metasurface can; (2) this proposed acoustic metasurface is compact and simple; (3) relatively easy to generate and low-cost.

I agree with the two latter points, this acoustic metasurface does have a high degree of practicality. However, I have doubts about the first point. As far as I know, there is no special evidence that previously proposed coiling-space/labyrinthine structures cannot be used for underwater sounds, as long as the metamaterial has sufficient acoustic impedance contrast with water, or the structure is specially designed (see e.g. a numerical investigation "An acoustic metasurface with simultaneous phase modulation and energy attenuation", *Modern Physics Letters B* 32.23 (2018): 1850276).

At the same time, for the underwater acoustic metasurfaces/metamaterials, there are many published research works on this field, e.g. "Broadband reflective metasurface for focusing underwater ultrasonic waves with linearly tunable focal length. *Applied Physics Letters* 108.16 (2016): 163502", "Thin Fresnel zone plate lenses for focusing underwater sound. *Applied Physics Letters* 107.1 (2015): 014103", "Broadband focusing of underwater sound using a transparent pentamode lens. *The Journal of the Acoustical Society of America* 141.6 (2017): 4408-4417", "Broadband solid cloak for underwater acoustics. *Physical Review B* 95.18 (2017) 180104", "Design and demonstration of an underwater acoustic carpet cloak. *Scientific reports* 7.1 (2017): 705". The author has neither highlighted the underwater acoustic metasurfaces/metamaterials (and some related references) in their title or the abstract, I don't think this is their main novelty and selling point.

I am very sorry to say that I will keep my attitude and not recommend its publication.

Reviewer #2:

Remarks to the Author:

The manuscript presents one of the very few metamaterial-based devices designed for underwater operation. The measurements match very well the simulation results, which proves that very large devices having surprisingly well-controlled micro-scale features can be fabricated in bulk. In addition, it is remarkable that the imaginary parts of the impedance and refractive index are as small as they are, which indicates relatively low loss. I am not aware of any other large scale metamaterial structure engineered so precisely to implement complex passive beamforming. I believe this work is very well done and the results are significant and of interest to a large community, in particular those doing beamforming for underwater applications and for ultrasound imaging. I therefore recommend publication in *Nature Communications*.

Reviewers' comments on the revised manuscript

Reviewer #1 (Remarks to the Author):

According to the reply, the authors claim their novelties are: (1) the previous acoustic metasurfaces based on generalized Snell's law's cannot be used for underwater sound, but this proposed acoustic metasurface can; (2) this proposed acoustic metasurface is compact and simple; (3) relatively easy to generate and low-cost.

I agree with the two latter points, this acoustic metasurface does have a high degree of practicality. However, I have doubts about the first point. As far as I know, there is no special evidence that previously proposed coiling-space/labyrinthine structures cannot be used for underwater sounds, as long as the metamaterial has sufficient acoustic impedance contrast with water, or the structure is specially designed (see e.g. a numerical investigation "An acoustic metasurface with simultaneous phase modulation and energy attenuation", *Modern Physics Letters B* 32.23 (2018): 1850276).

At the same time, for the underwater acoustic metasurfaces/metamaterials, there are many published research works on this field, e.g. "Broadband reflective metasurface for focusing underwater ultrasonic waves with linearly tunable focal length. *Applied Physics Letters* 108.16 (2016): 163502", "Thin Fresnel zone plate lenses for focusing underwater sound. *Applied Physics Letters* 107.1 (2015): 014103", "Broadband focusing of underwater sound using a transparent pentamode lens. *The Journal of the Acoustical Society of America* 141.6 (2017): 4408-4417", "Broadband solid cloak for underwater acoustics. *Physical Review B* 95.18 (2017) 180104", "Design and demonstration of an underwater acoustic carpet cloak. *Scientific reports* 7.1 (2017): 705". The author has neither highlighted the underwater acoustic metasurfaces/metamaterials (and some related references) in their title or the abstract, I don't think this is their main novelty and selling point.

I am very sorry to say that I will keep my attitude and not recommend its publication.

Reviewer #2 (Remarks to the Author):

The manuscript presents one of the very few metamaterial-based devices designed for underwater operation. The measurements match very well the simulation results, which proves that very large devices having surprisingly well-controlled micro-scale features can be fabricated in bulk. In addition, it is remarkable that the imaginary parts of the impedance and refractive index are as small as they are, which indicates relatively low loss. I am not aware of any other large scale metamaterial structure engineered so precisely to implement complex passive beamforming. I believe this work is very well done and the results are significant and of interest to a large community, in particular those doing beamforming for underwater applications and for ultrasound imaging. I therefore recommend publication in *Nature Communications*.

Authors' answers to the reviewers

Authors: First, we sincerely thank both reviewers for their careful reading of the revised manuscript and the supplementary information we provided for the first resubmission.

Answer to reviewer #1

Reviewer #1: *“According to the reply, the authors claim their novelties are: (1) the previous acoustic metasurfaces based on generalized Snell's law's cannot be used for underwater sound, but this proposed acoustic metasurface can; (2) this proposed acoustic metasurface is compact and simple; (3) relatively easy to generate and low-cost. I agree with the two latter points, this acoustic metasurface does have a high degree of practicality.”*

Authors: We are grateful to reviewer #1 for their careful reading of our documents and for their positive opinion of the “degree of practicality” of our metasurfaces, which is a very important point when designing and creating new functional wave devices.

Reviewer #1: *“However, I have doubts about the first point. As far as I know, there is no special evidence that previously proposed coiling-space/labyrinthine structures cannot be used for underwater sounds, as long as the metamaterial has sufficient acoustic impedance contrast with water, or the structure is specially designed (see e.g. a numerical investigation “An acoustic metasurface with simultaneous phase modulation and energy attenuation”, Modern Physics Letters B 32.23 (2018): 1850276).”*

Authors: We understand reviewer #1's concern about the coiling-space approach for underwater acoustics. In view of the successful results obtained in the past for shaping airborne-wave fronts with coiling-space metasurfaces, it is tempting to transpose this concept to waterborne waves. The main issue in this transposition to water as the propagating medium is recovery of the forced (guided) wave path through the labyrinthine structure. This can easily be done with airborne waves travelling between solid walls made of hard polymers or metals with an acoustic impedance Z (from 10^6 Pa.s/m for PDMS to $67 \cdot 10^6$ Pa.s/m for tungsten, which is the highest impedance available). These values are approximately 10,000 times higher than the acoustic impedance of air, 400 Pa.s/m (see the html link in the footnote for the acoustic properties of different media¹). With such a large acoustic impedance mismatch between the host medium and the solid boundaries of the labyrinthine path, it can be concluded that the walls are rigid (or have no wave motion within). However, this claim is formally/experimentally wrong (a certain amount of acoustic energy is always transferred from the host medium to the boundary/walls) and is just an approximation whose validity depends on the (very) strong impedance ratio. In the case of airborne metasurfaces designed using a coiling-space approach, the acoustic motion within the walls is so small compared to that of the air that it can be neglected and does not affect the main targeted acoustic features in the host medium.

¹ https://www.nde-ed.org/GeneralResources/MaterialProperties/UT/ut_matlprop_index.htm

When water is the host medium ($Z_{\text{water}} = 1.5 \cdot 10^6 \text{ Pa}\cdot\text{s}/\text{m}$), the impedance mismatch with labyrinthine walls decreases to the order of the unity for polymers (or to the order of 30 for metals¹). Thus, no material can be said to be “rigid” compared to water but simply “hard”. Since those figures are not comparable with those for airborne waves, it is a misconception to assume that an impedance ratio of, for example, 30 (for steel) is sufficiently high that the elasticity of the solid/hard medium can be neglected. Researchers in underwater acoustics/ultrasonics labs know that water acoustically couples extremely well to hard media and that thin structures made of hard materials are even quasi-transparent to long waves. This leads to a second issue: if the labyrinthine solid structure couples with water, guided wave generation (dynamics of the walls) or/and transmission from one cell to another through the elastic walls (unwilled transparency) can occur. Consequently, the elasticity of the walls impacts the targeted function of the metasurface. This impact can be quantified (theoretically and numerically) by considering the actual/physical case with appropriate mechanical parameters of the solid part.

Therefore, we anticipate that an efficient coiling-space metasurface for underwater acoustics is more difficult to design and fabricate than its airborne counterpart. Future attempts will investigate this.

In their comment, reviewer #1 refers to the work by Baozhu CHENG *et al.* (“An acoustic metasurface with simultaneous phase modulation and energy attenuation”, *Modern Physics Letters B* 32.23 (2018): 1850276) as a work demonstrating the possibility of a metasurface for underwater sound based on coiling space. A careful reading of that paper shows that the “rigid” part of the coiling-space metasurface for air was treated numerically with real material parameters (ABS plastic). However, for the underwater case, the protocol for the numerical treatment of the “rigid” walls was not clearly stated. By default, the main text suggests that the numerical simulation for water uses the same coiling-space structure as that for air: “rigid” walls made of ABS plastic. The trouble in this case is that the acoustic impedance of ABS plastic (mass density: 1,190 kg/m³; acoustic velocity: 1,360 m/s) is 1.1 times that of water. As a result, the metasurface should be quasi-transparent to incoming sound with no coiling-space effect. The pressure level in the “rigid” region of the simulation (Fig. 8b of the paper) is 0, as if the hard walls have been modelled numerically by setting the pressure level automatically to 0 in that specific region. However, a physical “elastic” labyrinthine structure cannot exhibit a zero pressure level within the solid part (especially when the acoustic impedances of the different components are similar), which contradicts the numerical results obtained in that paper. That is why we are not convinced by this approach and await experimental evidence.

Finally, notably, Haberman and Norris stated in a recent paper that “*the space coiling structure is not applicable for underwater devices because of the low contrast between the bulk moduli of common materials and water.*” (see “Broadband focusing of underwater sound using a transparent pentamode lens”, X. Su *et al.*, *The Journal of the Acoustical Society of America* **141**, 6, 4408-4417, June 2017).

Reviewer #1: “At the same time, for the underwater acoustic metasurfaces/metamaterials, there are many published research works on this field, e.g. “Broadband reflective metasurface for focusing underwater ultrasonic waves with linearly tunable focal length. *Applied Physics Letters* 108.16 (2016): 163502”, “Thin Fresnel zone plate lenses for focusing underwater sound. *Applied Physics Letters* 107.1 (2015): 014103”, “Broadband focusing of underwater sound using a transparent pentamode lens. *The Journal of the Acoustical Society of America* 141.6 (2017): 4408-4417”, “Broadband solid cloak for underwater acoustics. *Physical Review B* 95.18 (2017) 180104”, “Design and demonstration of an underwater acoustic carpet cloak. *Scientific reports* 7.1 (2017): 705”. The author has neither highlighted the underwater acoustic metasurfaces/metamaterials (and some related references) in their title or the abstract, I don’t think this is their main novelty and selling point.”

Authors: Our message to the metamaterials/metasurfaces community is not to claim that no relevant work has been done on the topic of metamaterial-based wave devices in water. We appreciate the above-listed papers (and others) as important research in the field of underwater meta-devices, which are much less studied and developed than their airborne wave analogues. However, we did not deliberately mix the concepts of *acoustic metasurfaces*, *acoustic metamaterials* and *phononic crystals*. Each of those concepts has been defined by the community according to specific geometric or/and physical features for the related devices. Even if all those concepts target the same acoustic functions, they generally involve different types of structures with different internal physical mechanisms. In this paper, we focus on the metasurface concept corresponding to flat devices with **sub-wavelength thickness** to control phase, amplitude, polarisation or any other wave property along a 2D surface. That is why our list of references preferentially highlights works dealing with very thin devices. Below, comments are provided for each reference provided by reviewer #1. Note that only the authors of the two first papers on this list focus on the metasurfaces concept.

1. “Broadband reflective metasurface for focusing underwater ultrasonic waves with linearly tunable focal length.” *Applied Physics Letters* 108.16 (2016): 163502.

This paper proposes gradient grooves on a brass substrate whose overall thickness is 5 mm. In the 0.45-0.55 MHz frequency band, the longest wavelength in water is 3.2 mm, which is smaller than the thickness of the structure. Therefore, it is not a sub-wavelength scale device and hardly meets the definition of a metasurface.

2. “Thin Fresnel zone plate lenses for focusing underwater sound.” *Applied Physics Letters* 107.1 (2015): 014103.

A Fresnel Zone Plate consists of a set of concentric rings that alternate between opaque and transparent. The incident wave will diffract around the opaque zones, and diffracted waves constructively interfere at a desired focal point. The radius of each ring follows the principle of constructive interference, leading to a monochromatic functionality. In contrast, our metasurface works over a broad frequency band because our porous polymers are very weakly dispersive.

3. “Broadband focusing of underwater sound using a transparent pentamode lens.” The Journal of the Acoustical Society of America 141.6 (2017): 4408-4417.

This is a GRIN phononic crystal lens whose thickness is much larger than the external wavelength because the achievable refractive index range is small. Therefore, this structure cannot be strictly considered an acoustic metasurface with a sub-wavelength thickness.

4. “Broadband solid cloak for underwater acoustics.” Physical Review B 95.18 (2017) 180104 and “Design and demonstration of an underwater acoustic carpet cloak.” Scientific reports 7.1 (2017): 705.

These two papers deal with cloaking devices, which are practically out of the scope of our present work. Nevertheless, as the design of our metasurfaces is based on the fine control of functionally graded material properties along the metasurface, the transformation acoustics used to design the cloak and the carpet cloak can be formally used in our design. For instance, our gradient-index flat lenses for focusing can be viewed as the transformation of a conventional convex or concave lens with constant acoustic index. Carpet cloaks based on metasurfaces have been recently proposed for EM waves and airborne waves, but a realization for underwater acoustics has not been reported.

Reviewer #1: *“I am very sorry to say that I will keep my attitude and not recommend its publication.”*

Authors: We hope to convince reviewer #1 that formulating and assembling porous elastomers with a tuneable acoustic index n in a very broad range ($n=c_{\text{water}}/c_{\text{porous}}$ from 1.5 to more than 25) provides a new approach for the design of functional materials for water ultrasonics, especially for sub-wavelength metasurfaces.

Answer to reviewer #2

Authors: We thank reviewer #2 for their careful reading and kind recommendation to publish our manuscript in *Nature Communications*.